# Increased Depression and Anxiety Disorders during the COVID-19 Pandemic in Children and Adolescents: A Literature Review

**DOI:** 10.3390/life11111188

**Published:** 2021-11-05

**Authors:** Justyna Śniadach, Sylwia Szymkowiak, Przemysław Osip, Napoleon Waszkiewicz

**Affiliations:** 1General Hospital in Kolno, 18-500 Kolno, Poland; 2HCP Medical Center in Poznań, 61-485 Poznań, Poland; szymkowiaksylwia@gmail.com (S.S.); osip1976@gmail.com (P.O.); 3Clinic of Psychiatry, Medical University of Bialystok, 15-089 Bialystok, Poland; napoleonwas@yahoo.com

**Keywords:** depression, anxiety disorders, children, adolescents, COVID-19, remote learning, mental health, mental well-being

## Abstract

Depression and anxiety disorders are a serious and increasingly commonly diagnosed problem at present. The problem applies not only to adults but also, increasingly often, to children and adolescents as well. The outbreak of the COVID-19 pandemic has further compounded the issue. There are still relatively few publications that show that quarantine and social isolation have a negative effect on the psychological well-being of children and adolescents. Above all, the situation applies to children and adolescents with pre-existing predispositions and to individuals suffering from mental disorders. The key factor in this situation seems to be putting the needs of young people first so that they can safely return to school. It is also important to provide them with effective treatment strategies and methods with which to deal with this stressful and potentially traumatic situation. Most of the mental health research during the COVID-19 pandemic has been conducted in Asia and Europe, where the disease first spread. This article presents an overview of the results of the latest Polish and international studies on the increase in depression and anxiety disorders among children and adolescents during the COVID-19 pandemic worldwide. It concludes with recommendations regarding mental health support for young people, and further directions for treatment are suggested.

## 1. Introduction

### 1.1. Depressive and Anxiety Disorders Worldwide

It has long been known that depression and anxiety disorders are serious diseases. According to the World Health Organization’s definition, the main symptoms of depression are sadness, feelings of guilt, loss of interests and pleasure, disturbed sleep and appetite, low self-esteem, fatigue, and decreased concentration. Depression has significant effects on quality of life and the ability to function in every area of life. Importantly, this illness is often accompanied by thoughts of self-harm or suicide. For many years, the frequency of mental disorders has been steadily growing, both among adults and among children and adolescents. Nowadays, more than 264 million people suffer from depression worldwide [1].

Generalized anxiety disorder (GAD) is a serious psychiatric condition, affecting up to 6% of the population during their lifetime. If it is not appropriately treated, it has a chronic course and carries a high burden of disability and for the public. Its manifestation is complicated by comorbidity with other psychiatric disorders, such as major depressive disorder (MDD), which may additionally aggravate the outcome and contribute to a poor response to treatment. It is estimated that 264 million adults around the globe suffer from anxiety [2].

### 1.2. Impact of the Pandemic on Mental Health

Worldwide mental health has been significantly impacted by the coronavirus pandemic. The years 2020 and 2021 continue to be marked by the COVID-19 pandemic. The coronavirus-related disease caused by the SARS-CoV-2 virus has been spreading worldwide for the last 1.5 years. According to WHO data, over 187 million cases have been diagnosed globally, including over 4 million fatalities [3]. Its many negative effects in the form of mental disorders are becoming more frequent, and this tendency looks set to continue in the coming months. Along with the next wave of the coronavirus, depression or anxiety will be the most frequent accompanying factors. According to a document from the World Health Organization entitled “The Mental Health Action Plan 2013–2020”, depression now accounts for 4.3% of the global burden of all diseases. It is also worth emphasizing that depression is on the list of the 20 most important causes of disability [4,5].

The latest studies carried out during the pandemic confirm the significant impact that the COVID-19 pandemic has had on the mental health of individuals, causing anxiety, depressive symptoms, anger, insomnia, denial, and fear [6]. Quarantine status has led to negative psychological consequences such as health anxiety, loneliness, and financial worry [7]. What is more, “headline stress disorder” has been observed during the pandemic. This disorder is characterized by strong emotional responses (such as stress and anxiety) to the endless media reports, and it may cause physical symptoms including insomnia and palpitations. This is why the COVID-19 pandemic can be interpreted, to all intents and purposes, as a collective traumatic event that has generated post-traumatic symptoms. From this perspective, research and clinical practice have shown that trauma per se is a powerful risk factor for mental disorders including post-traumatic stress disorder (PTSD) [7,8].

### 1.3. Impact of the Coronavirus Pandemic on the Mental Well-Being of Children and Adolescents

The COVID-19 pandemic has had a strong impact on the functioning of children and adolescents, in particular on their well-being and mental health. The time immediately following school closures was extremely difficult. It was related to the experience of a severe crisis among many young people [9]. It was a difficult and stressful experience for most young people. The life they knew thus far was radically changed almost overnight, and on many levels. This carried with it the risk that their basic needs would not be satisfied [10]. The mental health of children and adolescents during the COVID-19 pandemic is a very important issue in the field of public health, especially in terms of the potential, projected long-term effects [11].

Prior to the outbreak of the coronavirus pandemic, the worldwide frequency of mental disorders among children and adolescents was estimated to be 13.4% [12]. The COVID-19 pandemic has introduced a series of new sources of stress into the daily lives of children and adolescents. It is predicted that the above may contribute to mental illness, including depression, anxiety, and stress-related symptoms [13]. However, there are no uniform, up-to-date data on the degree by which the incidence of mental illness, including depression, in children and adolescents has increased between the outbreak of the pandemic and the present time. 

Since March 2020, researchers have been looking at how lockdowns and the coronavirus pandemic have affected mental health in young people, both in Poland and around the world. In this article, the latest available publications in this area are discussed in the context of the direct impact of SARS-CoV-2 on the human body and also of the social consequences of a pandemic, including isolation, stress, fear of disease, remote learning, or economic problems. Finally, we present our conclusions, and our directions for further proceedings are outlined.

## 2. Materials and Methods

Two independent reviewers screened titles and abstracts for relevance. A literature search was conducted in the PubMed, Google Scholar, and Scopus databases, as well as in other available sources. We included clinical studies, reviews, articles, meta-analyses, and case studies regarding increased incidence of depression and anxiety disorders during the COVID-19 pandemic in children and adolescents. The search strategy consisted of the following keywords: ‘depression’, ‘anxiety disorders’, ‘children’, ‘adolescents’, ‘COVID-19’, ‘remote learning’, ‘mental health’, and ‘mental well-being’ as well as combinations of these terms. We then excluded all articles that described young people over the age of 22 and adults in order to be able to analyze only data from children and adolescents up to 22 years of age. Relevant articles were then included with the intention to cover the widest possible spectrum of impact of SARS-CoV-2 on increased depression and anxiety disorders during the COVID-19 pandemic in children and adolescents. We conducted additional manual searches of the references of the related articles in order to gather information about the relevant supporting literature.

We restricted our search to studies published between January 2020 and September 2021. Only English-language articles were included. The search of the PubMed, Google Scholar, and Scopus databases, as well as other available sources, produced 140 articles, most of which derived from PubMed and all of whose titles and abstracts we read. We excluded 124 articles, which did not fulfil the search criteria. After reading the full texts, we considered 14 articles. Ten articles came from Pubmed, and another four were from other sources (Figure 1).

## 3. Literature Review: Results

### 3.1. Poland

In 2017, the Foundation for the Empowerment of Children (Empowering Children Foundation) published a report entitled “Children Count, 2017—A report on threats to the safety and development of children in Poland”. The research tool was an internet questionnaire filled in by children. From the report, it became clear that there was a systematic increase in the number of children being hospitalized in Poland owing to mental illness. The report also showed that Poland is second (in Europe) only to Germany in terms of the number of suicide attempts that result in death among children aged 10–19 years [14].

On the instructions of the Minister of Health, the Institute of Integrated Prophylaxis produced a report in April of 2020 entitled “Youth in the pandemic. For whom is it most difficult?”. The studied population comprised 2475 school-aged people (13–19 years). From the published report, it appears that every fourth subject felt bad or very bad. Various problems were cited by the subjects of the study: 78% said they missed meeting their friends and peers, and 63% reported feeling smothered by schoolwork, which they felt had increased with the introduction of remote learning. Every third subject said there was a tense atmosphere at home. Adolescents reported feelings of uncertainty, regression of social skills, and deregulation of home rhythms leading to effects on physical and mental health. Three quarters of class eight primary school pupils and 82% of final-year high school students reported feelings of fear related to their approaching exams. The frequency with which young people reported experiencing difficult emotions increased significantly. In particular, the respondents more often felt nervous and irritated, lacking in energy, lonely, anxious, and sad. One in five (21%) said the support they received from adults was insufficient for them [15].

The study “Remote learning and adaptation to social conditions during the coronavirus epidemic” was carried out in 34 primary and secondary schools from all over Poland, online, in the period from 12 May to 12 June 2020. The study was carried out using the diagnostic survey method. The research sample included the parents, teachers, and pupils of classes 6–8 of primary school and all secondary school classes (sample sizes: pupils *n* = 1284, parents *n* = 979, teachers *n* = 671).

All groups of respondents (students, parents, and teachers) agreed that their current mental and physical well-being was worse compared with the time before the pandemic. About 10% of the surveyed youth showed clear symptoms of depression: 9% of them felt sad all the time, 10% felt lonely and depressed all the time, and 9% of young people admitted that they wanted to cry all the time. Feelings of sadness, depression, willingness to cry, and loneliness in teachers and parents occurred much less frequently than in students (3–6% in the case of teachers and only 1–2% in the case of parents). Students and teachers who reported that important relationships in their lives had deteriorated during the pandemic experienced a greater impact on emotional symptoms and the risk of depression and psychosomatic problems. High levels of depressive mood were experienced by 17% of students, 13% of teachers, and 5% of parents. Analysis of the data on the general intensity of depressive moods of boys and girls showed clear gender differences: General high-level concerns were reported by 23% of girls and 8% of boys. Girls, more often than boys, declared that they experience sadness (12%), loneliness (14%), depression (13%), or the urge to cry all the time (12%). Among boys, the frequency of the same symptoms was in the range of 4–5%. Therefore, it can be concluded that the general mental well-being of girls was poorer than that of boys. Regarding the analysis of the data concerning the general intensity of the depression levels of primary and secondary school students, it should be noted that the general high level applies to 24% of secondary school students, 13% of primary school students, and 13% of students attending technical schools. Thus, it can be concluded that high school students suffered depression symptoms more frequently than others [16].

Another Polish study was conducted on the instructions of the “Schools with Class Foundation” in 2021. The study was conducted using an Internet questionnaire directed to the teachers (N-1535). This study showed that depression was the greatest problem among pupils, affecting as many as 40% of the studied group, though symptoms were most often noticed in the fifth month. The teachers also reported that depression was the biggest problem among secondary school students, with as many as 65% affected. Therefore, it can be clearly assumed that depressive mood and the related risk of suicide attempts among children and adolescents should be covered by urgent preventive and therapeutic measures during the pandemic and after students return to their educational institutions [17].

### 3.2. Germany

The number of cases of depression in children and adolescents in Germany has practically doubled over the past decade. According to an analysis of data from people insured in the German health insurance fund (KKH), a significant increase in other mental diseases has also been observed, and the COVID-19 pandemic, which has been going on since 2020, has further strengthened this phenomenon. The analysis of these data was made available by a German magazine on behalf of the KKH [18]. Data from 209,332 children and adolescents aged 6–18 years were analyzed. According to an analysis of insured individuals in 2019, around 12% of children and adolescents (25,000) were treated for mental illness. Extrapolation of this figure to all children and adolescents in Germany would amount to around 1.3 million.

Between 2009 and 2019, the number of cases of depression increased significantly in Germany (an increase of 97 percent). During this period, adaptation disorders (+72%), burnout disorders (+54.7%), anxiety disorders (+45%), and eating disorders (+13%) also increased significantly. At the same time, in no other federal state of Germany were there as many minors suffering from mental illness in 2019 as there were in Berlin. Almost 14 percent of children and adolescents aged 6–18 years were treated for mental disorders.

According to German health insurance, in the year 2020, from the time of the COVID-19 outbreak, the number of children and teenagers treated for eating disorders rose by about 60%. Other psychological disorders such as depression or burnout syndrome rose by around 30%.

The data available from Germany suggest an increase of more than 20% in all mental illnesses in children and adolescents aged from 6 to 18 years [18].

The above results are in line with another large-scale German study. COPSY was the first nationwide, representative German study to investigate the mental health and quality of life of children and adolescents during the pandemic. Interviews were conducted with *n* = 1586 parents of children and adolescents aged 7 to 17, of which *n* = 1040 were young people aged 11 to 17. The results were compared with data from a representative longitudinal study (BELLA) conducted before the pandemic. People aged 11 to 17 provided self-assessments, from 26 May to 10 June 2020. Data were analyzed using bivariate tests and descriptive statistics. The results showed that 71% of children and adolescents felt burdened by the first wave of the COVID-19 pandemic. Compared with the pre-pandemic period, subjects reported lower health-related quality of life, and the proportion of children and adolescents with mental health problems almost doubled while their health behavior deteriorated. Socially disadvantaged children felt particularly burdened by the COVID-19 pandemic [19].

### 3.3. Italy and Spain

Italy was the first European country to be affected by the coronavirus. Countermeasures taken by the Italian government to contain the spread of the virus consisted mainly of a nationwide quarantine and social distancing orders. School closures that followed destabilized the daily lives of millions of children and young people, representing around 16% of the Italian population. Accordingly, experts verbalized concerns about the psychological impact that the blockade and the pandemic could have on children and adolescents [20].

A study conducted in Italy and Spain in 2020 provided the first evidence of the negative impact of extended quarantine on the lives of children and adolescents in those countries. These two European countries were the hardest hit by the pandemic. The study conducted showed a deterioration of the emotional state and behavior of children, with difficulties related to concentration, irritability, loneliness, and boredom. There were some differences in the results, however. Spanish children seemed to be more deeply affected by symptoms than their Italian counterparts. It was the authors’ opinion that this difference arose from the fact that Spanish children were kept in complete isolation, while Italian children were allowed to take short walks, close to home. It may be noteworthy, however, that Italian parents more frequently reported that their children seemed sad than was the case in Spain. It is likely that this was because the quarantine in Italy lasted longer [21].

The deepest effects of quarantine and social isolation in Italy were felt by young people with serious mental conditions, such as psychoses, autism spectrum disorders, and anxiety. Children suffering from pre-existing mental illnesses or other conditions require more support to deal with uncertainties and to tolerate negative feelings. In such patients, a breakdown of routine or feelings of loneliness can worsen their feelings of well-being and their psychological state. As a result of closures of, or limited access to, support services, this group of patients and their therapeutic needs were not met with the necessary care and attention. At the time of the outbreak of the pandemic, most hospital services were given over to the fight against COVID-19 while other services, including mental health, were suspended (except in acute cases) [22].

### 3.4. China

According to studies carried out in China, it may be estimated that the COVID-19 outbreak took its toll on the physical, social, or mental health of 87% of pupils worldwide. There had been, however, no rigorous testing done at the time. 

A cross-sectional online survey of primary and secondary school pupils from Shanghai, China, was conducted in March 2020. The study sought to estimate the prevalence of symptoms of depression, anxiety, and stress, and the level of life satisfaction among children and adolescents experiencing home quarantine and confinement schools in Shanghai due to COVID-19. The internet survey was conducted among 4391 students from primary school (classes 1–5), middle school (classes 6–9), and high school (classes 10–12). The average age of the study group was 11.86 ± 2.32 years and ranged from 6 to 17 years. Girls represented 49% of the study group. Among the subjects, 24.9% showed symptoms of anxiety, 19.7% showed symptoms of depression, and 15.2% showed signs of stress. Five hundred (11.5%) of the subjects met criteria for depression, anxiety, and stress simultaneously. Symptoms were more severe and were most frequently reported in middle schools, and least frequently in primary schools. No differences were observed between boys and girls with regards to either frequency or severity of symptoms. It was observed, however, that symptoms of depression, anxiety, and stress were all greater among those who took a more negative approach to the home quarantine and who did not discuss COVID-19 with their parents [23]. 

Three of the newest reports from China showed an increase in depressive disorders as well as anxiety and conditions associated with psychological stress among children and adolescents during the COVID-19 pandemic [24,25,26].

A study carried out by Xie et al. (*n* = 2330) showed that as many as 22.6% of children reported some symptoms of depression based on a shortened form of the Childhood Depression Inventory. Meanwhile, 18.9% of the children reported symptoms in line with the Screen for Child Anxiety-Related Emotional Disorders [25].

Another large-scale study in China (*n* = 8079) of adolescents aged 12–18 years showed that the incidence of depression and anxiety symptoms was 43.7% and 37.4%, respectively [26].

Another study’s objective was to explore the impact of the COVID-19 pandemic on somatic symptoms among Chinese college and primary school students, in order to provide reference data pertaining to the mental health of this population in the context of a public health emergency. This study found the presence of somatic symptoms such as bodily pain and difficulty breathing, though only 2.4% of the children reported the symptoms. In the study (*n* = 209 primary school students (116 girls, 93 boys)) used a somatic self-rating scale. Among the entire cohort, concern regarding COVID-19 was positively correlated with the occurrence of somatic symptoms. Somatic symptoms were more likely among college students expressing greater concern regarding the threat to life and health posed by COVID-19, and the efficacy of prevention and control measures [24]. 

Another study from China provided parental reports of symptoms of mental illness seen in their children. In the study (*n* = 320), typical symptoms of depressive disorders were reported according to the DSM-5 classification. Reported symptoms include attachment disorder (37%), inattention (33%), irritability (32%), and worry (28%). Other symptoms included the fear of death in the family (22%), sleep disturbances (22%), poor appetite (18%), fatigue (17%), nightmares (14%), and discomfort/agitation (13%) [27].

Regarding risk factors for mental illness, the results regarding age [25,26] indicated more severe symptoms of mental illness in older children, one study found inconclusive results in relation to age and gender [27], while a third showed no statistically significant influence of age on the severity of mental health issues in children and adolescents [24]. Regarding the gender of children, American and Chinese studies [26,28] have shown that female gender is a risk factor for higher rates of depression and anxiety symptoms, while one study [25] showed that gender is not a determinant for symptoms of anxiety or depression.

Studies by Jiao et al. and Xie et al. confirmed that symptoms of mental illness were higher in children living in areas with higher infection rates [26,27].

Two studies presented evidence that fear of infection and perceived risk to life were associated with poorer mental health outcomes [24,25]. With regards to protective factors, the study by Zhou et al. showed that awareness of COVID-19 protects against depression and anxiety symptoms [26]. Similarly, the research of Jiao et al. showed that reading, exercise, and media entertainment can help reduce the mental stress of children related to COVID-19 [27].

### 3.5. The United States

The coronavirus pandemic brought chaos to the lives of people the world over. The United States has become one of the most affected countries, with the highest number of infections and deaths related to COVID-19 up to April 2020 [29]. By March 2020, many steps had been taken to reduce the spread, including home quarantines, bans on interstate travel, remote working, online education, and others. 

Studies carried out in the United States (*n* = 303 parents of which 45% had children under the age of 10 years) found that 40.1% of parents reported signs that their children were suffering. Meanwhile, 6.3% reported uncertainty regarding their children’s health, and 30.9% reported no suffering. Additionally, this report showed that parents with greater anxiety more frequently reported suffering among their children. It was also observed that greater symptoms of anxiety and depression in children and adolescents were related to financial stress in the family [30].

Another study looked at changes in symptoms of depression and anxiety since the pre-pandemic period compared to the peak of the pandemic in spring 2020. The study sample included adolescents and young adults (*n* = 451, ages 12–22) living in Long Island, New York, the early epicenter of COVID-19 in the USA. Symptoms of depression (Childhood Depression Inventory) and anxiety symptoms (Screen for Child Anxiety Related Disorders) were assessed between December 2014 and July 2019. Subsequently, after the COVID-19 outbreak, symptoms were reassessed between March 27 and May 15, 2020. According to the publication, adolescents and young adults in the early period and in the epicenter of the COVID-19 pandemic in the USA experienced a significant increase in symptoms of depression and anxiety. This trend was most noticeable among women. These fears were probably related to forced social isolation, both at school and at home. The results showed that the COVID-19 pandemic had a negative impact on the mental health of adolescents [31].

## 4. Discussion

The rapid spread of COVID-19 during the present pandemic has posed one of the most significant global challenges in recent years. 

There are currently very few publications on depressive and anxiety disorders in children and adolescents during the COVID-19 pandemic. Meanwhile, the available publications are alarming and clearly indicate a significant increase in depression and anxiety symptoms in children and adolescents. It is difficult to properly estimate how much these issues have intensified because of COVID-19 as there are no unambiguous pre-COVID-19 data. The studies conducted so far are few, cover various groups of risk factors, and were carried out using various methodologies, which further compounds the problem regarding comparability. There is a particular lack of research in younger children. Thus far, it has also been impossible to conduct long-term longitudinal studies to estimate and assess whether the severity of depressive and anxiety disorders in children and adolescents will persist over time, after the pandemic is over, or whether it will decrease or even increase further.

As mentioned at the beginning of this article, the worldwide incidence of mental disorders amongst children and adolescents, prior to the outbreak of the coronavirus pandemic, was estimated to be 13.4%.

Studies carried out in Poland have demonstrated an increase in depression and anxiety disorders up to 26.6% (~2–4-fold higher) [17]. The situation looks serious as well in Germany. Data available from the year 2020, gathered nationwide, show an increase of more than 20% in all mental illnesses in children and adolescents aged from 6 to 18 years (~2–3-fold higher) [18].

Studies carried out in Italy and Spain in 2020 also show evidence of the negative impact of the pandemic on the mental health of children and adolescents, although there is a lack of concrete data [20,21,22]. Some data from China [25] show increasing symptoms of depression in children and adolescents, with increases of about 9.2% and 6.4% in symptoms of anxiety (~1–2-fold higher). Another study shows increasing symptoms of depressive disorders by about 30% and an increase in anxiety disorders by 24% (~3–4-fold higher) [26]. Studies carried out in the USA [30] showed that 40.1% of parents reported signs of mental suffering in their children (such as depression and anxiety), which gives an increase of 26.7% (~3–4-fold higher). These results are shown in Table 1.

It is worth mentioning that the majority of mental health disorders begin in childhood, making it essential that mental health needs are identified early and treated during this sensitive period of child development. If untreated, mental health problems can lead to many negative health and social outcomes [11].

The COVID-19 pandemic could worsen existing mental health problems and lead to more cases among children and adolescents due to the unique combination of a public health crisis, social isolation, and economic recession. The economic downturn is linked to increased mental health problems in adolescents, which could be related to the way the economic downturn affects adult unemployment, adult mental health, and child abuse [32].

Teachers, administrators, and policy makers must minimize the disruption that the closure of schools causes to academic development. Schools offer students many other critical services besides education. For example, schools are the main source of food for many students, and food security is a serious loss during school closures. However, other services are just as important for children’s health and need to be addressed.

Meanwhile, college and university students are stressed by the evacuation of dormitories and the cancellation of anticipated events such as exchange studies and graduation ceremonies. Some have lost their part-time jobs as local businesses shut down. Final year students are concerned about the labor market they will soon enter. Students are more at risk than we think, especially with the current academic and financial burden.

One potentially overlooked role played by schools is the delivery of health care, and in particular mental health services. Schools have long served as the de facto mental health system for many children and adolescents [33].

Social distancing measures can result in social isolation in a violent home, and abuse can increase in times of economic insecurity and stress. Increased rates of child abuse, neglect, and abuse were also noted in previous public health emergencies, such as the Ebola outbreak in West Africa in 2014–2016.

However, little is known about the long-term effects of large-scale disease outbreaks on the mental health of children and adolescents. While there is some research on the psychological effects of severe acute respiratory syndrome (SARS) on patients and health care professionals, little is known about the effects on ordinary citizens. The evidence is especially scarce in children and adolescents. This is an important research gap. COVID-19 is much more widespread than SARS and other epidemics on a global scale. As the pandemic continues, it is important to support children and young people struggling with bereavement and problems related to parental unemployment or loss of household income. There is also a need for long-term mental health monitoring of young people and investigation of how prolonged school closures, stringent social distancing measures, and the pandemic itself affect the well-being of children and adolescents [13].

It seems particularly important to support young people who have had the worst experiences during the period of online education and who are, above all, people with special educational needs. It seems crucial to organize distance learning in such a way that it does not cause additional difficulties and to minimize the experienced stress [34].

During school closures and lockdowns, the priority in educational and supportive activities should be to ensure that basic psychological needs are met (primarily the need for security) and to build and maintain supportive relationships in the educational and home environments [35].

Equally, it is important to properly prepare educational systems for the return of children to school, and to create a proper support system in educational institutions and external institutions. This may include psychological and pedagogical counseling centers, community psychological and psychotherapeutic support centers, family support centers, and others [10].

## 5. Limitations and Future Directions

Some important limitations should be noted. At the time of writing, there was still very limited information regarding the increased incidence of depression and anxiety disorders during the COVID-19 pandemic in children and adolescents. The available data concern a few countries, although it is known that the coronavirus pandemic has affected practically the entire world.

Another potential limitation is that most of the studies included in this review were based on data collected during the early phase of the outbreak. As COVID-19 is rapidly evolving, these early estimates may change as more information is collected. We believe that additional research using case reports from other countries would be extremely useful since different demographic and cultural characteristics of the population may play an important role in determining outbreak trajectories and clinical outcomes at the population level.

The fourth wave of the coronavirus largely affects children and adolescents. Therefore, more and more publications on this subject might be expected. It is highly recommended that another systematic review should be carried out in the near future, including additional data.

Previous publications were mainly concerned with the impact of the restrictions, social distancing, or school closures on the mental health of children. The first articles on the impact of COVID-19 disease on physical health and related secondary mental well-being (including depressive and anxiety disorders) among children and adolescents will probably be published shortly.

Another variable that may be considered in the future is the increasingly common vaccination of children and adolescents. At present, there are no data on how the vaccine affects the mental health of children.

## 6. Conclusions

The results of this review show that there are clear gaps in the literature regarding the impact of the COVID-19 pandemic on the mental health of children and adolescents around the world. It would be advisable to conduct further studies in many countries, using strong methodology with detailed reporting, and giving a clear assessment of the mental well-being of children and adolescents both before and after the COVID-19 pandemic, using cohort studies [36]. Longitudinal studies in more countries are recommended. It would also be useful to carry out further systematic reviews in order to be able to assess over time how the coronavirus pandemic has affected the mental well-being of children and adolescents, from its beginning to its end. It is also advisable to monitor these factors over the years following the end of the pandemic.

## Figures and Tables

**Figure 1 life-11-01188-f001:**
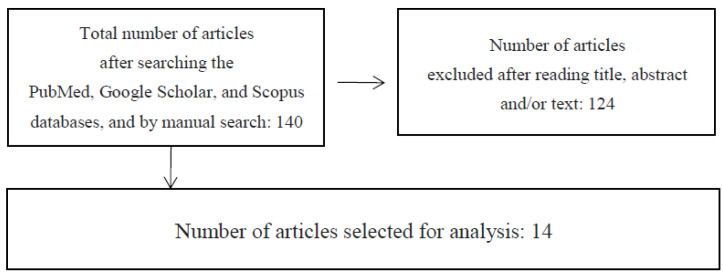
Flow chart summarizing the search for articles concerning depression and anxiety disorders during the COVID-19 pandemic in children and adolescents.

**Table 1 life-11-01188-t001:** Study characteristics.

Study	Country	Participants	Mental Health Outcomes	Mental Health Measures	Notes
The Institute of Integrated Prophylaxis, 2020	Poland	2475 people aged 13–19 years	fear, nervous, irritated, lacking in energy, lonely, anxious, sad	internet survey	-
Ptaszek et al., 2020	Poland	pupils *n* = 1284, parents *n* = 979, teachers *n* = 671	sadness, loneliness, depression	internet survey	- the general mental well-being of girls was poorer than that of boys - high school students suffered depression symptoms more frequently than others
Schools with Class Foundation, 2021	Poland	teachers *n* = 1535	depression, risk of suicide	internet survey with closed and open questions	older students are more likely to experience depression
Mittagsmagazin, 2020	Germany	209,332 children and adolescent aged 6–18 years	depression, adaptation disorders, burnout disorders, anxiety disorders, eating disorders	internet survey	increase of more than 20% in all mental illnesses in children and adolescents aged from 6 to 18 years
Ravens-Sieber et al., 2020	Germany	1586 parents	psychosomatic symptoms: irritability, sleeping problems, headache, feeling low, stomach ache	- The KIDSCREEN Questionnaires—Quality of life questionnaires for children and adolescents - Strengths and Difficulties Questionnaire - Screen for Child Anxiety Related Emotional Disorders - Center for Epidemiological Studies Depression Scale for Children - Patient Health Questionnaire	socially disadvantaged children felt particularly burdened by the COVID-19 pandemic
Orgiles et al., 2020	Italy and Spain	1143 parents of Italian and Spanish children aged 3–18 years	difficulty concentrating, boredom, irritability, restlessness, nervousness, feelings of loneliness, uneasiness, worries	internet survey	children suffering from pre-existing mental illnesses or other conditions require more support to deal with uncertainties and to tolerate negative feelings
Tang et al., 2021	China	4391 people aged 6–17 years	anxiety, depression, stress	internet survey	- symptoms were more severe and were most frequently reported in middle schools - no differences were observed between boys and girls
Liu et al., 2020	China	*n* = 209 (primary school students (116 girls, 93 boys))	somatic symptoms, anxiety, depression	somatic self-rating scale	- somatic symptoms were more likely among college students expressing greater concern regarding the threat to life and health posed by COVID-19 - no statistically significant influence of age on the severity of mental health issues in children and adolescents
Xie et al., 2020	China	2330 students	depression, anxiety	- Children’s Depression Inventory Short Form - Screen for Child Anxiety Related Emotional Disorders	- more severe symptoms of mental illness in older children - gender is not determinant for symptoms of anxiety or depression
Zhou et al., 2020	China	8079 people aged 12–18 years	depression, anxiety	-PHQ-9 -GAD-7	- more severe symptoms of mental illness in older children - female gender is a risk factor for higher rates of depression and anxiety symptoms
Jiao et al., 2020	China	320 children (168 girls, 142 boys) aged 3–18	discomfort and agitation, nightmares, fatigue, poor appetite, sleeping disorders, fear for the health of relatives, obsessive request for updates, worry, irritability, inattention, clinginess	questionnaire was completed online by parents and incorporated DSM-5 criteria	- inconclusive results due to age and gender - symptoms of mental illness were higher in children living in areas with higher infection rates
Oosterhoff et al., 2020	USA	657 adolescents (13–18 years)	anxiety, depression	Patient-Reported Outcomes Measurement Information System (PROMIS): anxiety and depression scales	being female was a risk factor for higher rates of depressive and anxiety symptoms
Rosen et al., 2020	USA	303 parents, of which 45% had children under the age of 10 years (mean age: 43 years)	distress	researchers created questions about observation of child distress	greater symptoms of anxiety and depression in children and adolescents were related to financial stress in the family
Hawes et al., 2021	USA	*n* = 451, ages 12–22	depression, anxiety	- Childhood Depression Inventory - Screen for Child Anxiety Related Disorders	female gender is a risk factor for higher rates of depression and anxiety symptoms

## Data Availability

Not applicable.

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
