# Peer review of "Increased Depression and Anxiety Disorders during the COVID-19 Pandemic in Children and Adolescents: A Literature Review"

_life, 2021, doi:10.3390/life11111188_

Round 1

Reviewer 1 Report

Dear colleagues, I hope this message find you well.

Thank you for giving me the opportunity of reading the work “Increased depression and anxiety disorders during the COVID19 pandemic in children and adolescents- literature review, it has been a very big pleasure to collaborate reviewing this manuscript. The topic of this paper is very interesting and it seems necessary to delve it. However, there are several questions to improve before to publish it: 

Introduction

  • I recommended to divide the introduction into several subsections in order to do it more intuitive.
  • On the other hand, when you explain the signs of psychological distress and mental health as a result of COVID-19 (page 1), it is necessary to add more data. I recommend you to add this paper recently published which has been develop in Italy: https://doi.org/10.3390/ijerph18147422
  • Objectives are not clear.

Method

  • I recommend adding more information. For example, you could add a figure detailing the whole process.

Discussion

  • It is necessary to describe in more detail the practical and theoretical implications.
  • Limitations should be explained more deeply.

Other issues

  • It is advisable to do a proofreading.Extensive editing of English language and style are required.

Author Response

Dear reviewer,

Thanks a lot for reviewing my manuscript. It was my pleasure reading your feedback and comments.

I have corrected my article in line with your comments.

Introduction

As you recommended, I divided the introduction into several sections to make it more intuitive in reception.

In explaining the signs of mental stress from COVID-19, I added more data to keep the goals of the article clear. Following your recommendation, I added some information from an article recently published in Italy ( https://doi.org/10.3390/ijerph18147422).

Methods

I added more information to the description of the methodology and included a figure describing the entire process.

Discussion

I have described the practical and theoretical implications in more detail. I explained the limitations in more detail.

Other issues

I did a language correction. In addition, I asked my friend who is a native speaker to check and amend the article to make it is correct and free from linguistic errors.

I am very pleased to be able to resend my manuscript with the corrections I made. I sincerely hope that it now meets the requirements of the journal. I tried my best to apply to all the comments I received.

Best regards,

Justyna Śniadach

Reviewer 2 Report

It is an interesting and well-done study, on time, deals with very important problems now, but references from Poland should be translated into English in brackets.

Author Response

Dear reviewer,

Thanks a lot for reviewing my manuscript. It was my pleasure reading your feedback and comments.

I have corrected my article in line with your comments. I translated references from Poland  into English in brackets. I did a language correction. In addition, I asked my friend, who is a native speaker, to check and amend the article to make it correct and free from linguistic errors.

I am very pleased to be able to resend my manuscript with the corrections I made. I sincerely hope that it now meets the requirements of the journal. I tried my best to apply to all the comments I received.

Best regards,

Justyna Śniadach

Reviewer 3 Report

The reviewed paper deals with the increasing number of depression and anxiety disorders during the COVID-19 pandemic in children and adolescents. Young people suffer more than adults from the new situation because their basic needs are not fulfilled. The authors of this study did a literature review and present findings from different countries (e.g., Poland, Germany, and China). The results of the cited literature are well presented with descriptive statistics (e.g., percentages of subgroups). In the discussion section, the authors sum up their findings and interpretations.

I really enjoyed reading this article and thank the authors for the opportunity to review their interesting paper. In my opinion, this paper deals with a very important and timely topic and is well structured. The introduction guides the reader smoothly into this topic, all necessary background information (i.e., citing appropriate literature) is given. It was a pleasure to read these sections!

However, I see several issues with the ending of this paper. What I really missed was a table with key findings across the different countries. What is similar between the analyzed countries? What are the main differences? Moreover, I missed a limitation section before the conclusions. What are the most important limitations of this literature review? Did the authors cite all relevant literature in this field? Indeed, the authors report in the conclusion section that there are gaps in the literature on the impact of the COVID-19 pandemic (see page 7, line 335). Which gaps? The authors should explain this in more detail. Furthermore, I urge the authors to present at least one or two ideas of “strong methodology” (page 7, line 337) that should be taken into account when conducting future studies. Otherwise, this last sentence is a “black box” to the reader.

Besides these major issues, I detected some smaller errors and mis-spellings (note, this list is not complete). Hence, I have some comments/suggestions that I hope will help the authors to further develop this line of work:

  1. Introduction (page 2, line 68): Please delete the additional space in this sentence (see “We included  clinical studies […][“).
  2. Introduction (page 2, line 78): Please delete the unnecessary hyphen in the word “poss-sible”.
  3. Introduction (page 2, line 90): Please delete the additional space in this sentence (see “aged 10-19 years  [9]”.
  4. Literature review (page 3, line 133): Please delete the unnecessary dot in this sentence (see “[…] conducted in 2020. Are […]”
  5. Literature review (page 4, line 170): Please delete the additional space in this sentence (see “[…] health insurance,  in the year […]”
  6. Literature review (page 5, line 252): Please delete the unnecessary dot and space in this sentence (see “[…] irritability (32%) and  worry (28%)..”
  7. Literature review (page 6, line 271): “The coronavirus pandemic brought chaos to the lives of people the world over.” This sentence sounds awkward. Please re-arrange this sentence. The whole paper should be checked carefully before it is re-submitted. Perhaps the authors should ask a native speaker for checking their manuscript.
  8. Discussion (page 7, line 311): Please correct this sentence and add a second square bracket after the citation: “[…] through 12% [10, up to […]”

Author Response

Dear reviewer,

Thanks a lot for reviewing my manuscript. It was my pleasure reading your feedback and comments.

I have corrected my article in line with your comments. I corrected and developed the ending of my article. I have added a table including all the countries compared so that it is easier to compare the results and spot differences and similarities.

I've added a limitation section as per your recommendations.

I also added the concept of a "strong methodology" that should be taken into account when conducting future research.

In addition, I did a language correction. What is more I asked my friend, who is a native speaker, to check and amend the article to make it correct and free from linguistic errors.

I am very pleased to be able to resend my manuscript with the corrections I made. I sincerely hope that it now meets the requirements of the journal. I tried my best to apply to all the comments I received.

Best regards,

Justyna Śniadach

Round 2

Reviewer 1 Report

Congratulations! Your work has been improved significantly.

Author Response

Dear reviewer,

It is very nice to read such messages.I tried very hard to improve my manuscript.

It will be a great honor for me if my article is published in such a respected journal. Thanks for all your comments.

Best regards,

Justyna Sniadach

Reviewer 3 Report

I thank the authors for submitting a revised version of their manuscript entitled “Increased depression and anxiety disorders during the COVID‐19 pandemic in children and adolescents- literature review”.

The authors did a great job, improved their manuscript according to my suggestions, and responded to all my questions adequately. In my opinion, the revised manuscript increased a lot in comparison to the first version. Everything read more smoothly and I could follow authors’ argumentations easily. The authors re-wrote the discussion and the limitation section.

However, I detected two typos that should be corrected before the publication of this paper:

  • Title of this manuscript (page 1, line 3): Not “Increased depression and anxiety disorders during the COVID‐19 pandemic in children and adolescents- literature review” but rather “Increased depression and anxiety disorders during the COVID‐19 pandemic in children and adolescents- a literature review”. Please correct the title of this manuscript and add the word “a” before “literature review”.
  • Table 1 (page 9, line 390): The content of the presented Table 1 is great! However, Table 1 is being cut on the right side. Please format the table so that it fits on the page and nothing is lost.
  • Limitation and Future Directions (page 12, line 459): Not “[…] that another systematic review be carried out in the near future […]” but rather “[…] that another systematic review will be carried out in the near future […]”. Please correct this sentence and check the whole manuscript for other mistakes, especially in the added yellow sections.

Author Response

Dear reviewer,

It is very nice to read such messages.

I tried very hard to improve my manuscript.

I did all the changes you recomend. It will be a great honor for me if my article is published in such a respected journal.

Thanks for all your comments,

Justyna Sniadach